# A Similarity-preserving Neural Network Trained on Transformed Images Recapitulates Salient Features of the Fly Motion Detection Circuit

**Yanis Bahroun** [†]     **Anirvan M. Sengupta** [†‡]     **Dmitri B. Chklovskii**[†*]
[†]Flatiron Institute    [‡]Rutgers University    [*]NYU Langone Medical Center
{ybahroun,dchklovskii}@flatironinstitute.org,    anirvans@physics.rutgers.edu,

## Abstract

Learning to detect content-independent transformations from data is one of the central problems in biological and artificial intelligence. An example of such problem is unsupervised learning of a visual motion detector from pairs of consecutive video frames. Rao and Ruderman formulated this problem in terms of learning infinitesimal transformation operators (Lie group generators) via minimizing image reconstruction error. Unfortunately, it is difficult to map their model onto a biologically plausible neural network (NN) with local learning rules. Here we propose a biologically plausible model of motion detection. We also adopt the transformation-operator approach but, instead of reconstruction-error minimization, start with a similarity-preserving objective function. An online algorithm that optimizes such an objective function naturally maps onto an NN with biologically plausible learning rules. The trained NN recapitulates major features of the well-studied motion detector in the fly. In particular, it is consistent with the experimental observation that local motion detectors combine information from at least three adjacent pixels, something that contradicts the celebrated Hassenstein-Reichardt model.

## 1  Introduction

Humans can recognize objects, such as human faces, even when presented at various distances, from various angles and under various illumination conditions. Whereas the brain performs such a task almost effortlessly, this is a challenging unsupervised learning problem. Because the number of training views for any given face is limited, such transformations must be learned from data comprising different faces, or in a content-independent manner. Therefore, learning content-independent transformations plays a central role in reverse engineering the brain and building artificial intelligence.

Perhaps the simplest example of this task is learning a visual motion detector, which computes the optic flow from pairs of consecutive video frames regardless of their content. Motion detector learning was addressed by Rao and Ruderman [31] who formulated this problem as learning infinitesimal translation operators (or generators of the translation Lie group). They learned a motion detector by minimizing, for each pair of consecutive video frames, the squared mismatch between the observed variation in pixel intensity values and that predicted by the scaled infinitesimal translation operator. Whereas such an approach learns the operators and evaluates transformation magnitudes correctly [31, 22, 42], its biological implementation has been lacking (see below).

The non-biological nature of the neural networks (NNs) derived from the reconstruction approach has been previously encountered in the context of discovery of latent degrees of freedom, e.g. dimensionality reduction and sparse coding [8, 26]. When such NNs are derived from the reconstruction-error-minimization objective they require non-local learning rules, which are not biologically plausible. To

overcome this, [28, 29, 30] proposed deriving NNs from objectives that strive to preserve similarity between pairs of inputs in corresponding outputs.

Inspired by [29, 30], we propose a similarity-preserving objective for learning infinitesimal translation operators. Instead of preserving similarity of input pairs as was done for dimensionality reduction NNs, our objective function preserves the similarity of input features formed by the outer product of variation in pixel intensity and pixel intensity which are suggested by the translation-operator formalism. Such objective is optimized by an online algorithm that maps onto a biologically plausible NN. After training the similarity-preserving NN on one-dimensional (1D) and two-dimensional (2D) translations, we obtain an NN that recapitulates salient features of the fly motion detection circuit.

Thus, our main contribution is the derivation of a biologically plausible NN for learning content-independent transformations by similarity preservation of outer product input features.

## 1.1 Contrasting reconstruction and similarity-preservation NNs

We start by reviewing the NNs for discovery of latent degrees of freedom from principled objective functions. Although these NNs do not detect transformations, they provide a useful analogy that will be important for understanding our approach. First, we explain why the NNs derived from minimizing the reconstruction error lack biological plausibility. Then, we show how the NNs derived from similarity preservation objectives solve this problem.

To introduce our notation, the input to the NN is a set of vectors, $\mathbf{x}_t \in \mathbb{R}^n, t = 1, \ldots, T$, with components represented by the activity of $n$ upstream neurons at time, $t$. In response, the NN outputs an activity vector, $\mathbf{y}_t \in \mathbb{R}^m, t = 1, \ldots, T$, where $m$ is the number of output neurons.

The reconstruction approach starts with minimizing the squared reconstruction error:

$$\min_{\mathbf{W}, \mathbf{y}_{t=1\ldots T} \in \mathbb{R}^m} \sum_t ||\mathbf{x}_t - \mathbf{W}\mathbf{y}_t||^2 = \min_{\mathbf{W}, \mathbf{y}_{t=1\ldots T} \in \mathbb{R}^m} \sum_{t=1}^T \left[ ||\mathbf{x}_t||^2 - 2\mathbf{x}_t^\top \mathbf{W}\mathbf{y}_t + \mathbf{y}_t^\top \mathbf{W}^\top \mathbf{W}\mathbf{y}_t \right], \quad (1)$$

possibly subject to additional constraints on the latent variables $\mathbf{y}_t$ or on the weights $\mathbf{W} \in \mathbb{R}^{n \times m}$. Without additional constraints, this objective is optimized offline by a projection onto the principal subspace of the input data, of which PCA is a special case [24].

In an online setting, the objective can be optimized by alternating minimization [26]. After the arrival of data sample, $\mathbf{x}_t$: firstly, the objective (1) is minimized with respect to the output, $\mathbf{y}_t$, while the weights, $\mathbf{W}$, are kept fixed, secondly, the weights are updated according to the following learning rule derived by a gradient descent with respect to $\mathbf{W}$ for fixed $\mathbf{y}_t$:

$$\dot{\mathbf{y}}_t = \mathbf{W}_{t-1}^\top \mathbf{x}_t - \mathbf{W}_{t-1}^\top \mathbf{W}_{t-1} \mathbf{y}_t, \qquad \mathbf{W}_t \longleftarrow \mathbf{W}_{t-1} + \eta \left( \mathbf{x}_t - \mathbf{W}_{t-1}\mathbf{y}_t \right) \mathbf{y}_t^\top, \qquad (2)$$

In the NN implementations of the algorithm (2), the elements of matrix $\mathbf{W}$ are represented by synaptic weights and principal components by the activities of output neurons $y_j$, Fig. 1a [23].

However, implementing update (2)right in the single-layer NN architecture, Fig. 1a, requires non-local learning rules making it biologically implausible. Indeed, the last term in (2)right implies that updating the weight of a synapse requires the knowledge of output activities of all other neurons which are not available to the synapse. Moreover, the matrix of lateral connection weights, $-\mathbf{W}_{t-1}^\top \mathbf{W}_{t-1}$, in the last term of (2)left is computed as a Gramian of feedforward weights; a non-local operation. This problem is not limited to PCA and arises in nonlinear NNs as well [26, 18].

Whereas NNs with local learning rules have been proposed [26] their two-layer feedback architecture is not consistent with most biological sensory systems with the exception of olfaction [17]. Most importantly, such feedback architecture seems inappropriate for motion detection which requires speedy processing of streamed stimuli.

To address these difficulties, [29] derived NNs from similarity-preserving objectives. Such objectives require that similar input pairs, $\mathbf{x}_t$ and $\mathbf{x}_{t'}$, evoke similar output pairs, $\mathbf{y}_t$ and $\mathbf{y}_{t'}$. If the similarity of a pair of vectors is quantified by their scalar product, one such objective is similarity matching (SM):

$$\min_{\forall t \in \{1, \ldots, T\}: \, \mathbf{y}_t \in \mathbb{R}^m} \frac{1}{2} \sum_{t, t'=1}^T \left( \mathbf{x}_t \cdot \mathbf{x}_{t'} - \mathbf{y}_t \cdot \mathbf{y}_{t'} \right)^2. \qquad (3)$$

This offline optimization problem is also solved by projecting the input data onto the principal subspace [44, 5, 19]. Remarkably, the optimization problem (3) can be converted algebraically to a tractable form by introducing variables $\mathbf{W}$ and $\mathbf{M}$ [30]:

$$\min_{\{\mathbf{y}_t \in \mathbb{R}^m\}_{t=1}^T} \min_{\mathbf{W} \in \mathbb{R}^{n \times m}} \max_{\mathbf{M} \in \mathbb{R}^{m \times m}} \left[ \sum_{t=1}^T \left(-2\mathbf{x}_t^\top \mathbf{W} \mathbf{y}_t + \mathbf{y}_t^\top \mathbf{M} \mathbf{y}_t\right) + T \operatorname{Tr}(\mathbf{W}^\top \mathbf{W}) - \frac{T}{2} \operatorname{Tr}(\mathbf{M}^\top \mathbf{M})\right]. \quad (4)$$

In the online setting, first, we minimize (4) with respect to the output variables, $\mathbf{y}_t$, by gradient descent while keeping $\mathbf{W}, \mathbf{M}$ fixed [29]:

$$\dot{\mathbf{y}}_t = \mathbf{W}^\top \mathbf{x}_t - \mathbf{M} \mathbf{y}_t. \quad (5)$$

To find $\mathbf{y}_t$ after presenting the corresponding input, $\mathbf{x}_t$, (5) is iterated until convergence. After the convergence of $\mathbf{y}_t$, we update $\mathbf{W}$ and $\mathbf{M}$ by gradient descent and gradient ascent respectively [29]:

$$W_{ij} \leftarrow W_{ij} + \eta \left(x_i y_j - W_{ij}\right), \qquad M_{ij} \leftarrow M_{ij} + \eta \left(y_i y_j - M_{ij}\right). \quad (6)$$

Algorithm (5), (6) can be implemented by a biologically plausible NN, Fig. 1b. As before, activity (firing rate) of the upstream neurons encodes input variables, $\mathbf{x}_t$. Output variables, $\mathbf{y}_t$, are computed by the dynamics of activity (5) in a single layer of neurons. The elements of matrices $\mathbf{W}$ and $\mathbf{M}$ are represented by the weights of synapses in feedforward and lateral connections respectively. The learning rules (6) are local, i.e. the weight update, $\Delta W_{ij}$, for the synapse between $i^{\text{th}}$ input neuron and $j^{\text{th}}$ output neuron depends only on the activities, $x_i$, of $i^{\text{th}}$ input neuron and, $y_j$, of $j^{\text{th}}$ output neuron, and the synaptic weight. Learning rules (6) for synaptic weights $\mathbf{W}$ and $-\mathbf{M}$ (here minus indicates inhibitory synapses, see Eq.(5)) are Hebbian and anti-Hebbian respectively.

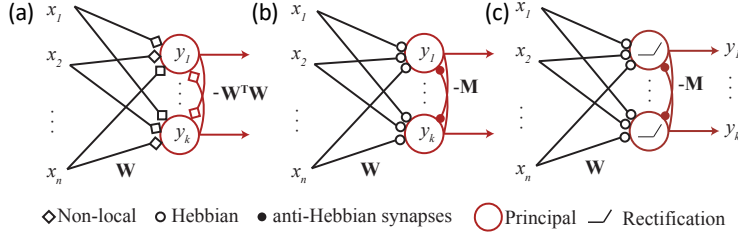

◇Non-local   o Hebbian   • anti-Hebbian synapses   ◯Principal   ⟋ Rectification

Figure 1: Single-layer NNs performing online (a) reconstruction error minimization (1) [23, 26], (b) similarity matching (SM) (3) [29], and (c) nonnegative similarity matching (NSM) (7) [28].

We now compare the objective functions of the two approaches. After dropping invariant terms, the reconstructive objective function has the following interactions among input and output variables: $-2\mathbf{x}_t^\top \mathbf{W} \mathbf{y}_t + \mathbf{y}_t^\top \mathbf{W}^\top \mathbf{W} \mathbf{y}_t$ (Eq 1). The SM approach leads to $-2\mathbf{x}_t^\top \mathbf{W} \mathbf{y}_t + \mathbf{y}_t^\top \mathbf{M} \mathbf{y}_t$, ( Eq 4). The term linear in $\mathbf{y}_t$, a cross-term between inputs and outputs, $-2\mathbf{x}_t^\top \mathbf{W} \mathbf{y}_t$, is common in both approaches and is responsible for projecting the data onto the principal subspace via the feedforward connections in Fig.1ab. The terms quadratic in $\mathbf{y}_t$'s decorrelate different output channels via a competition implemented by the lateral connections in Fig.1ab and are different in the two approaches. In particular, the inhibitory interaction between neuronal activities $y_j$ in the reconstruction approach depends upon $\mathbf{W}^\top \mathbf{W}$, which is tied to trained $\mathbf{W}$ in a non-local way. In contrast, in the SM approach the inhibitory interaction matrix $\mathbf{M}$ is learned for $y_j$'s via a local anti-Hebbian rule.

The SM approach can be applied to other computational tasks such as clustering and learning manifolds by tiling them with localized receptive fields [34]. To this end we modify the offline optimization problem (3) by constraining the output, $\mathbf{y}_t \in \mathbb{R}_+^m$, which represents assignment indices (as e.g. in the K-means algorithm):

$$\min_{\forall t \in \{1, \ldots, T\}: \mathbf{y}_t \in \mathbb{R}_+^m} \frac{1}{2} \sum_{t, t'=1}^T \left(\mathbf{x}_t \cdot \mathbf{x}_{t'} - \mathbf{y}_t \cdot \mathbf{y}_{t'}\right)^2. \quad (7)$$

Such nonnegative SM (NSM), just like the optimization problem (3), (7) can be converted algebraically to a tractable form by introducing similar variables $\mathbf{W}$ and $\mathbf{M}$ [28]. The synaptic weight update rules presented in (6) remain unchanged and the only difference between the online solutions of (3) and (7) is the dynamics of neurons which, instead of being linear, are now rectifying, Fig. 1c.

In the next section, we will address transformation learning. Similarly, we will review the reconstruction approach, identify the key term analogous to the cross-term $-2\mathbf{x}_t^\top \mathbf{W}\mathbf{y}_t$, and then alter the objective function, so that the cross-term is preserved but the inhibition between output neurons can be learned in a biologically plausible manner.

## 2 Learning a motion detector using similarity preservation

Now, we focus on learning to detect transformations from pairs of consecutive video frames, $\mathbf{x}_t$, and $\mathbf{x}_{t+1}$. We start with the observation that much of the change in pixel intensities in consecutive frames arises from a translation of the image. For infinitesimal translations, pixel intensity change is given by a linear operator (or matrix), denoted by $\mathbf{A}^a$, multiplying the vector of pixel intensity scaled by the magnitude of translation, denoted by $\theta^a$. Because for a 2D image multiple directions of translation are possible, there is a set of translation matrices with corresponding magnitudes. Our goal is to learn both the translation matrices from pairs of consecutive video frames and compute the magnitudes of translations for each pair. Such a learning problem will reduce to the one discussed in the previous section, but performed on an unusual feature – the outer product of pixel intensity and variation of pixel intensity vectors.

### 2.1 Reconstruction-based transformation learning

We represent a video frame at time, $t$, by the pixel intensity vector, $\mathbf{x}_t$, formed by reshaping an image matrix into a vector. For infinitesimal transformations, the difference, $\Delta\mathbf{x}_t$, between two consecutive frames, $\mathbf{x}_t$ and $\mathbf{x}_{t+1}$ is:

$$\Delta\mathbf{x}_t = \mathbf{x}_{t+1} - \mathbf{x}_t = \sum_{a=1}^{K} \theta_t^a \mathbf{A}^a \mathbf{x}_t \quad , \quad \forall t \in \{1, \ldots, T-1\}. \tag{8}$$

where, for each transformation, $a \in \{1, \ldots K\}$, between the frames, $t$ and $t+1$, we define a transformation matrix $\mathbf{A}^a$ and a magnitude of transformations, $\theta_t^a$. Whereas for image translation $\mathbf{A}^a$ is known to implement a spatial derivative operator, we are interested in learning $\mathbf{A}^a$ from data in unsupervised fashion.

Previously, unsupervised algorithms for learning both $\mathbf{A}^a$ and $\theta_t^a$ were derived by minimizing with respect to $\mathbf{A}^a$ and $\theta_t^a$ the prediction-error squared [31] where optimal $\mathbf{A}^a$ and $\theta_t^a$ minimize the mismatch between the actual image and the one computed based on the learned model:

$$\sum_{t} \|\Delta\mathbf{x}_t - \sum_{a=1}^{K} \theta_t^a \mathbf{A}^a \mathbf{x}_t\|^2 = \sum_{t} \left[ \|\Delta\mathbf{x}_t\|^2 - 2\Delta\mathbf{x}_t^\top \sum_{a=1}^{K} \theta_t^a \mathbf{A}^a \mathbf{x}_t + \| \sum_{a=1}^{K} \theta_t^a \mathbf{A}^a \mathbf{x}_t\|^2 \right]. \tag{9}$$

Whereas solving (9) in the offline setting leads to reasonable estimates of $\mathbf{A}^a$ and $\theta_t^a$ [31], it is rather non-biological. In a biologically plausible online setting the data are streamed sequentially and $\theta_t^a$ ($\mathbf{A}^a$) must be computed (updated) with minimum latency. The algorithm can store only the latest pair of images and a small number of variables, i.e. sufficient statistic, but not any significant part of the dataset. Although a sketch of neural architecture was proposed in [31], it is clear from Section 1.1 that due to the quadratic term in the output, $\theta_t^a$, a detailed architecture will suffer from the same non-locality as the reconstruction approach to latent variable NNs (1).

As the cross-term in (9) plays a key role in projecting the data (Section 1.1), we re-write it as follows:

$$\sum_{t} \Delta\mathbf{x}_t^\top \sum_{a=1}^{K} \theta_t^a \mathbf{A}^a \mathbf{x}_t = \sum_{i,j,t,a} \Delta\mathbf{x}_{t,i} \theta_t^a \mathbf{A}^a_{i,j} \mathbf{x}_{t,j} = \sum_{i,j,t,a} \theta_t^a \mathbf{A}^a_{i,j} \Delta\mathbf{x}_{t,i} \mathbf{x}_{t,j} = \sum_{t} \Theta_t \boldsymbol{A} \mathrm{Vec}(\Delta\mathbf{x}_t \mathbf{x}_t^\top), \tag{10}$$

where we introduced $\boldsymbol{A} \in \mathbb{R}^{K \times n^2}$, the matrix whose components represents the vectorized version of the generators, $\boldsymbol{A}_{a,:} = \mathrm{Vec}(\mathbf{A}^a), \forall a \in \{1, \ldots, K\}$ and $\Theta_t = (\theta_t^{a=\{1\ldots K\}})^\top$, the vector whose components represent the magnitude of the transformation, $a$, at time, $t$.

Eq. (10) shows that the cross-term favors aligning $\boldsymbol{A}_{a,:}$ in the direction of the outer product of pixel intensity variation and pixel intensity vectors, $\mathrm{Vec}(\Delta\mathbf{x}\mathbf{x}^\top)$. Although central to the learning of transformations in (9), the outer product of pixel intensity variation and pixel intensity vectors was not explicitly highlighted in the transformation-operator learning approach [31, 10, 22].

## 2.2 Why the outer product of pixel intensity variation and pixel intensity vectors?

Here, we provide intuitions for using outer products in content-independent detection of translations. For simplicity, we consider 1D motion in a 1D world. Motion detection relies on a correspondence between consecutive video frames, $\mathbf{x}_t$ and $\mathbf{x}_{t+1}$.

One may think that such correspondences can be detected by a neuron adding up responses of the displaced filters applied to $\mathbf{x}_t$ and $\mathbf{x}_{t+1}$. While possible in principle, such neuron's response would be highly dependent on the image content [20, 21]. This is because summing the outputs of the two filters amounts to applying an OR operation to them which does not selectively respond to translation.

To avoid such dependence on the content, [20] proposed to invoke an AND operation, which is implemented by multiplication. Specifically, consider forming an outer product of $\mathbf{x}_t$ and $\mathbf{x}_{t+1}$ and summing its values along each diagonal. If the image is static then the main diagonal produces the highest correlation. If the image is shifted by one pixel between the frames then the first sub(super)-diagonal yields the highest correlation. If the image is shifted by two pixels - the second sub(super)-diagonal yields the highest correlation and so on. Then, if the sum over each diagonal is represented by a different neuron, the velocity of the object is given by the most active neuron. Other models relying on multiplications are "mapping units" [15], "dynamic mappings" [41] and other bilinear models [25].

Our algorithm for motion detection adopts multiplication to detect correspondences but computes an outer product between the vectors of pixel intensity, $\mathbf{x}_t$, and pixel intensity *variation*, $\Delta\mathbf{x}_t$. Compared to the approach in [20], one advantage of our approach is that we do not require separate neurons to represent different velocities but rather have a single output neuron (for each direction of motion), whose activity increases with velocity. Previously, a similar outer product feature was proposed in [3] (for a formal connection - see Supplement **A**). Another advantage of our approach is a derivation from the principled SM objective motivated by the transformation-operator formalism.

## 2.3 A novel similarity matching objective for learning transformations

Having identified the cross-term in (9) analogous to that in (1), we propose a novel objective function where the inhibition between output neurons is learned in a biologically plausible manner. By analogy with (Eq.3), we substitute the reconstruction-error-minimization objective by an SM objective for transformation learning. We denote the vectorized outer product between $\Delta\mathbf{x}_t$ and $\mathbf{x}_t$ as $\chi_t \in \mathbb{R}^{n^2}$:

$$\chi_{t,\alpha} = (\Delta\mathbf{x}_t\mathbf{x}_t^\top)_{i,j}, \quad \text{with } \alpha = (i-1)n+j, \tag{11}$$

We concatenate these vectors into a matrix, $\boldsymbol{\chi} \equiv [\chi_1, \ldots, \chi_T]$, as well as the transformation magnitude vectors, $\boldsymbol{\Theta} \equiv [\Theta_1, \ldots, \Theta_T]$. Using these notations, we introduce the following SM objective:

$$\min_{\boldsymbol{\Theta}\in\mathbb{R}^{K\times T}} \|\boldsymbol{\chi}^\top\boldsymbol{\chi} - \boldsymbol{\Theta}^\top\boldsymbol{\Theta}\|_F^2 = \min_{\Theta_1,\ldots,\Theta_T} \frac{1}{T^2}\sum_t^T\sum_{t'}^T(\chi_t^\top\chi_{t'} - \Theta_t^\top\Theta_{t'})^2. \tag{12}$$

To reconcile (9) and (12), we first show that the cross-terms are the same by introducing the following optimization over a matrix, $\mathbf{W} \in \mathbb{R}^{K\times n^2}$ as:

$$\frac{1}{T^2}\sum_{t=1}^T\sum_{t'=1}^T\Theta_t^\top\Theta_{t'}\chi_t^\top\chi_{t'} = \frac{1}{T^2}\sum_{t=1}^T\Theta_t^\top\left[\sum_{t'=1}^T\Theta_{t'}\chi_{t'}^\top\right]\chi_t = \max_{\mathbf{W}}\frac{2}{T}\sum_{t=1}^T\Theta_t^\top\mathbf{W}\chi_t - \text{Tr}\mathbf{W}^\top\mathbf{W} \tag{13}$$

Therefore, the SM approach yields the cross-term, $\Theta_t^\top\mathbf{W}\chi_t$ which is the same as $\Theta_t^\top\boldsymbol{A}\text{Vec}(\Delta\mathbf{x}_t\mathbf{x}_t^\top)$ in [31]. We can thus identify the rows $\mathbf{W}_{a,:}$ with the vectorized transformation matrices, $\text{Vec}(\mathbf{A}^a)$, Fig. 2a. Solutions of (12) are known to be projections onto the principal subspace of $\boldsymbol{\chi}$, the vectorized outer product of $\Delta\mathbf{x}_t$ and $\mathbf{x}_t$ which are equivalent, up to an orthogonal rotation, to PCA.

If we constrain the output to be nonnegative (NSM):

$$\min_{\boldsymbol{\Theta}\in\mathbb{R}_+^{K\times T}} \|\boldsymbol{\chi}^\top\boldsymbol{\chi} - \boldsymbol{\Theta}^\top\boldsymbol{\Theta}\|_F^2. \tag{14}$$

then by analogy with Sec. 1.1 [28], this objective function clusters data or tiles data manifolds [34].

## 2.4 Online algorithm and NN

To derive online learning algorithms for (12) and (14) we follow the similarity matching approach [29]. The optimality condition of each online problem is given by [28, 29] for SM and NSM respectively:

$$\text{SM:} \quad \Theta_t^* = \mathbf{W}\chi_t - \mathbf{M}\Theta_t^* \quad ; \quad \text{NSM:} \quad \Theta_t^* = \max(\mathbf{W}\chi_t - \mathbf{M}\Theta_t^*, 0) \quad , \tag{15}$$

with $\mathbf{W}$ and $\mathbf{M}$ found using recursive formulations, $\forall a \in \{1, \ldots, K\}, \forall \alpha \in \{1, \ldots, n^2\}$:

$$\mathbf{W}_{a\alpha} \leftarrow \mathbf{W}_{a\alpha} + \left( \Theta_{t-1,a}(\chi_{t-1,\alpha} - \mathbf{W}_{a\alpha}\Theta_{t-1,a}) \Big/ \hat{\Theta}_{t,a} \right) \tag{16}$$

$$\mathbf{M}_{aa' \neq a} \leftarrow \mathbf{M}_{aa'} + \left( \Theta_{t-1,a}(\Theta_{t-1,a'} - \mathbf{M}_{aa'}\Theta_{t-1,a}) \Big/ \hat{\Theta}_{t,a} \right) \tag{17}$$

$$\hat{\Theta}_{t,a} = \hat{\Theta}_{t-1,a} + (\Theta_{t-1,a})^2 \quad . \tag{18}$$

This algorithm is similar to the model proposed in [29], but it is more difficult to implement in a biologically plausible way. This is because $\chi_t$ is an outer product of input data and cannot be identified with the inputs to a single neuron. To implement this algorithm, we break up $\mathbf{W}$ into rank-1 components, each of which is computed in a separate neuron such that:

$$\Theta_{t,a}^* = \sum_i \Delta \mathbf{x}_{t,i} \sum_j W_{ija} \mathbf{x}_{t,j} - \sum_{a'} \mathbf{M}_{aa'} \Theta_{t,a'}^* \quad . \tag{19}$$

Each element of the tensor, $W_{ija}$ will be encoded in the weight of a feedforward synapse from the $j$-th pixel onto $i$-th neuron encoding $a$-th transformation (see Fig. 2a). Biologically plausible implementations of this algorithm are given in Section 3.

## 2.5 Numerical experiments

Here, we implement the biologically plausible algorithms presented in the previous subsection and report the learned transformation matrices. To validate the results of SM and NSM applied to the outer-product feature, $\chi$, we compare them with those of PCA and K-means, respectively, also applied to $\chi$ as formally defined in in Supplement **B**. These standard but biologically implausible algorithms were chosen because they perform similar computations in the context of latent variable discovery.

The 1D visual world is represented by a continuous profile of light intensity as a function of one coordinate. A 1D eye measures light intensity in a 1D window consisting of $n$ discrete pixels. To imitate self-motion, such window can move left and right by a fraction of a pixel at each time step. For the purpose of evaluating the proposed algorithms and derived NNs, we generated artificial training data by subjecting a randomly generated 1D image (Gaussian, exponentially correlated noise) to known horizontal subpixel translations. Then, we spatially whitened the discrete images by using the ZCA whitening technique [2].

We start by learning $K = 2$ transformation matrices using each algorithm. After the rows of the synaptic weights, W, are reshaped into $n \times n$ matrices, they can be identified with the transformation operators, $\mathbf{A}$. Then the magnitude of the transformation given by $\Delta \mathbf{x}_t^\top \mathbf{A} \mathbf{x}_t$, Fig. 2a.

**SM and PCA.** The filters learned from SM are shown in Fig.2c and those learned from PCA - in Fig.2e. The left panels of Fig.2ce represent the singular vectors capturing the maximum variance. They replicate the known operator of translation, a spatial derivative, found in [31]. The right panels of Fig.2ce show the singular vector capturing the second largest variance, which do not account for a known transformation matrix. In the absence of a nonnegativity constraint a reversal of translation is represented by a change of sign of the transformation magnitude.

**NSM and K-means.** The filters learned by NSM are shown in Fig.2d and those learned by K-means - in Fig. 2f. They are similar to the first singular vector learned by SM, PCA and [31]. However, in NSM and K-means the output must be nonnegative, so representing the opposite directions of motion requires two filters, which are sign inversions of each other.

For the various models, the rows of the learned operators, $\mathbf{A}^a$, are identical except for a shift, i.e. the same operator is applied at each image location. As expected, the learned filters compute a spatial derivative of the pixel intensity, red rectangle in Fig.2a. The learned weights can be approximated by

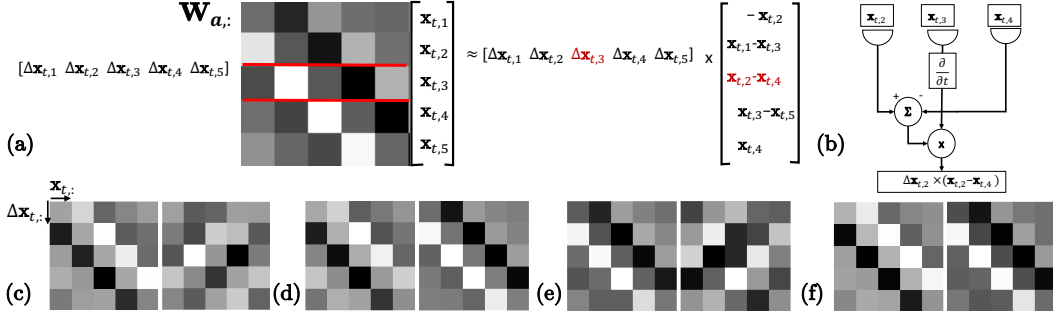

Figure 2: The rows of the synaptic weight matrix $\mathbf{W}_{a,:}$ are reshaped into $n \times n$ transformation matrices $\mathbf{A}^a$. Then, the magnitude of the transformation is $\Delta \mathbf{x}_t^\top \mathbf{A}^a \mathbf{x}_t$. Such a computation can be approximated by the cartoon model (b). Synaptic weights learned from 1D translation on a vector of size 5 pixels by (c) SM, (d) NSM, (e) PCA (decreasing eigenvalues), and (f) K-means.

the filter keeping only the three central pixels, Fig.2 which we name the cartoon model of the motion detector. It computes a correlation between the spatial derivative denoted by $\Delta_i \mathbf{x}_{t,i}$ and the temporal derivative, $\Delta_t \mathbf{x}_t$. Such algorithm may be viewed as a Bayesian optimal estimate of velocity in the low SNR regime (Supplement **C**) appropriate for the fly visual system[36].

The results presented in Fig.2 were obtained with $n = 5$ pixels, but the same structure was observed with larger values of $n$. Similar results were also obtained with models trained on moving periodic sine-wave gratings often used in fly experiments.

We also trained our NN on motion in the four cardinal directions, and planar rotations of two-dimensional images as was done in [31] and showed that our model can learn such transformations. By using NSM we can again distinguish between motion in the four cardinal directions, and clockwise and counterclockwise rotations, which was not possible with prior approaches (see Supplement **D**).

## 3 Learning transformations in a biologically plausible way

In this section, we propose two biologically plausible implementations of a motion detector by taking advantage of the decomposition of the outer product feature matrix into single-row components (19). The first implementation models computation in a mammalian neuron such as a cortical pyramidal cell. The second models computation in a Drosophila motion-detecting neuron T4 (same arguments apply to T5). In the following, for simplicity we focus on the cartoon model Fig.2b.

### 3.1 Multi-compartment neuron model

Mammalian neurons can implement motion computation by representing each row of the transformation matrix, W, in a different dendritic branch originating from the soma (cell body). Each such branch forms a compartment with its own membrane potential [14, 37] allowing it to perform its own non-linear computation the results of which are then summed in the soma. Each dendrite compartment receives pixel intensity variation from only one pixel via a proximal shunting inhibitory synapse [40, 16] and the pixel intensity vector via more distal synapses, Fig. 3a. We assume that the conductance of the shunting inhibitory synapse decreases with the variation in pixel intensity. The weights of the more distal synapses represent the corresponding row of the outer product feature matrix. When the variation in pixel intensity is low, the shunting inhibition vetoes other post-synaptic currents. When the variation in pixel intensity is high, the shunting is absent and the remaining post-synaptic currents flow into the soma. A formal analysis shows that this operation can be viewed as a multiplication [40, 16]. Different compartments compute such products for variation in intensity of different pixels, after which these products are summed in the soma (19), Fig. 3a.

The weight of a distal synapse is updated using a Hebbian learning rule applied to the corresponding pixel intensity available pre-synaptically and the transformation magnitude modulated by the shunting inhibition representing pixel intensity variation, Fig. 3b. The transformation magnitude is computed in the soma and reaches distal synapses via backpropagating dendritic spikes [38]. Such backpropagating signal is modulated by the shunting inhibition, thus implementing multiplication of the transformation

magnitude and pixel intensity variation (16), Fig. 3b . Competition between the neurons detecting motion in different directions is mediated by inhibitory interneurons [27].

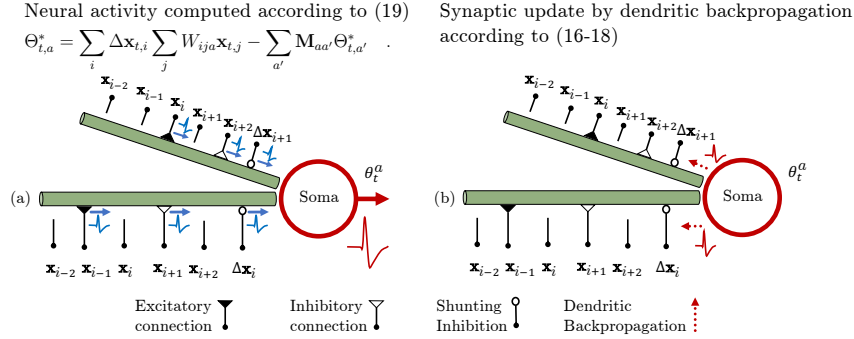

Figure 3: A multi-compartment model of a mammalian neuron. (a) Each dendrite multiplies pixel intensity variation signaled by the shunting inhibitory synapse and the weighted vector of pixel intensities carried by more distal synapses. Products computed in each dendrite are summed in the soma to yield transformation magnitude encoded in the spike rate. (b) Synaptic weights are updated by the product of the corresponding pre-synaptic pixel intensities and the backpropagating spikes modulated by the shunting inhibition.

## 3.2   A learned similarity preserving NN replicates the structure of the fly motion detector

The Drosophila visual system comprises retinotopically organized layers of neurons, meaning that nearby columns process photoreceptor signals (identified with $\mathbf{x}_i$ below) from nearby locations in the visual field. Unlike the implementation in the previous subsection, motion computation is performed across multiple neurons. The local motion signal is first computed in each of the hundreds of T4 neurons that jointly tile the visual field. Their outputs are integrated by the downstream giant tangential neurons. Each T4 neuron receives light intensity variation from only one pixel via synapses from neurons Mi1 and Tm3 and light intensities from nearby pixels via synapses from neurons Mi4 and Mi9 (with opposite signs) [39], Fig. 3c. Therefore, in each T4 neuron $\Delta x$ is a scalar and $\mathbf{W}$ is a vector and local motion velocity can be computed by a single-compartment neuron. If the weights of synapses from Mi4 and Mi9 of different columns represent $\mathbf{W}$, then the multiplication of $\Delta x$ and $\mathbf{W}\mathbf{x}$ can be accomplished as before using shunting inhibition. Competition among T4s detecting different directions of motion is implemented by inhibitory lateral connections.

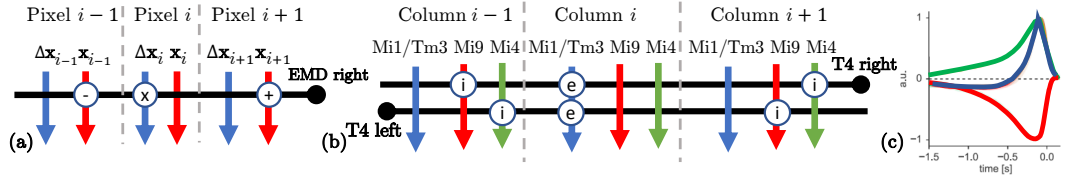

Figure 4: An NN trained on 1D translations recapitulates the motion detection circuit in Drosophila. (a) Each motion-detecting neuron receives pixel intensity variation signal from pixel $i$ and pixel intensity signals at least from pixels $i-1$ and $i+1$ (with opposite signs). (b) In Drosophila, each retinotopically organized column contains neurons Mi1/Tm3, Mi9, and Mi4 [39] which respond to light intensity in the corresponding pixel according to the impulse responses shown in (c) (from [1]). Each T4 neuron selectively samples different inputs from different columns [39]: it receives light intensity variation via Mi1/Tm3 and light intensity via Mi4 and Mi9 (with opposite signs).

Our model correlates inputs from *at least three* pixels in agreement with recent experimental results [39, 1, 11, 33], instead of *two* in the celebrated Hassenstein-Reichardt detector (HRD)[32]. In the fly, outputs of T4s are summed over the visual field in downstream neurons. The summed output of our detectors is equivalent to the summed output of HRDs and thus consistent with multiple prior behavioral experiments and physiological recordings from downstream neurons (see Supplement **E**).

There is experimental evidence for both nonlinear interactions of T4 inputs [33, 13] supporting a multiplicative model but also for the linear summation of inputs [11, 43]. Even if summation is linear, the neuronal output nonlinearity can generate multiplicative terms for outer product computation.

The main difference between our learned model (Fig.2a) and most published models is that the motion detector is learned from data using biologically plausible learning rules in an unsupervised setting. Thus, our model can generate somewhat different receptive fields for different natural image statistics such as that in ON and OFF pathways potentially accounting for minor differences reported between T4 and T5 circuits [39].

A recent model from [33] also uses inputs from three differently preprocessed inputs. Unlike our model that relies on a derivative computation in the middle pixel, the model in [33] is composed of a shared non-delay line flanked by two delay lines.

As shown in Supplement **E**, after integration over the visual field, the global signal from our cartoon model Fig.2b is equivalent to that from HRD. Same observation has been made for the model in [33]. Yet, the predicted output of a single motion detector in our model is different from both HRD and [33].

### 3.3 Experimentally established properties of the global motion detector

Until recently, most experiments confirmed the predictions of the HRD model. However, almost all of these experiments measured either the activity of downstream giant neurons integrating T4 output over the whole visual field or the behavioral response generated by these giant neurons. Because after integration over the visual field, the global signal from our cartoon model Fig.2b is equivalent to that from HRD, various experimental confirmations of the HRD predictions are inherited by our model. Below, we list some of the confirmed predictions.

**Dependence of the output on the image contrast.** Because HRD multiplies signals from the two photoreceptors its output should be quadratic in the stimulus contrast. Similarly, in our model, the output should be proportional to contrast squared because it is given by the covariance between time and space derivatives of the light intensity Supplement **C** each proportional to contrast. Note that this prediction differs from [31] whose output is contrast-independent. Several experiments have confirmed these predictions in the low SNR regime [12, 7, 9, 35, 4]. Of course, the output cannot grow unabated and, in the high SNR regime, the output becomes contrast independent. A likely cause is the signal normalization between photoreceptors and T4 [12].

**Oscillations in the motion signal locked to the visual stimulus.** In accordance with the oscillating output of HRD in response to moving periodic stimulus, physiological recordings have reported such phase-locked oscillations [6]. Our model reproduces such oscillations.

**Dependence of the peak velocity on the wavelength.** In our model, just like in the HRD, output first increases with the velocity of the visual stimulus and then decreases. The optimal velocity is proportional to the spatial wavelength of the visual stimulus because then the temporal frequency of the optimal stimulus is a constant given by the inverse of the time delay in one of the arms.

**In conclusion,** we learn transformation matrices using a similarity-preserving approach leading to a biologically plausible model of a motion detector. Generalizing our work to the learning of other content-preserving transformation will open a path towards principled biologically plausible object recognition.

### Acknowledgments

We are grateful to P. Gunn, and A. Genkin for discussion and comments on this manuscript. We thank D. Clark, J. Fitzgerald, E. Hunsicker, and B. Olshausen for helpful discussions.

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
