[Supplementary Material]

## Supplementary Materials

This is the supplementary material for NeurIPS 2019 paper: "A Similarity-preserving Neural Network Trained on Transformed Images Recapitulates Salient Features of the Fly Motion Detection Circuit" by Y. Bahroun, A. M. Sengupta, and D. B. Chklovksii.

## A  Alternative features for learning transformations

As described in Section 2.2 (main text), our algorithm for motion detection adopts multiplication to detect correspondences, it computes an outer product between the vectors of pixel intensity, $\mathbf{x}_t$, and pixel intensity variation, $\Delta\mathbf{x}_t$. Although an outer product feature was proposed in [3], our model most resembles that of [1]. We show in the following that in the first order our model and that of [1] are related.

Let us consider a mid-point in time between $t$ and $t+1$, denoted by $t+1/2$. Using the same approximation as before, $\mathbf{x}_{t+1/2}$ can be expressed as function of either $\mathbf{x}_t$ or $\mathbf{x}_{t+1}$ as

$$\mathbf{x}_{t+1/2} = \mathbf{x}_t + \frac{1}{2}\theta_t \mathbf{A}^a \mathbf{x}_t \tag{1}$$

$$= \mathbf{x}_{t+1} - \frac{1}{2}\theta_t \mathbf{A}^a \mathbf{x}_{t+1} \tag{2}$$

By subtracting the two equations above we obtain

$$\mathbf{x}_{t+1} - \mathbf{x}_t = \frac{1}{2}\theta_t \mathbf{A}^a (\mathbf{x}_{t+1} + \mathbf{x}_t). \tag{3}$$

We can thus rephrase the reconstruction error as:

$$\min_{\theta_t, \mathbf{A}^a} \sum_t \left\| \Delta\mathbf{x}_t - \frac{1}{2}\sum_{a=1}^{K} \theta_t^a \mathbf{A}^a (\mathbf{x}_{t+1} + \mathbf{x}_t) \right\|^2 \tag{4}$$

leading to the cross-term $\theta_t \mathbf{A}^a (\mathbf{x}_{t+1} + \mathbf{x}_t)\Delta\mathbf{x}_t^\top$, which corresponds to one proposed by [1]. An interesting observation regarding the outer product $(\mathbf{x}_{t+1} + \mathbf{x}_t)\Delta\mathbf{x}_t^\top$, is that, unlike the one used in the main text, this feature has a time-reversal anti-symmetry. This helps detecting the direction of change.

Another feature considered in [1] is

$$\mathbf{x}_{t+1}\mathbf{x}_t^\top - \mathbf{x}_t\mathbf{x}_{t+1}^\top \tag{5}$$

which exhibits an anti-symmetry both in time and in indices.

The anti-symmetric property of the learnable transformation operators is expected to arise from the data rather than by construction. In fact, only elements of the algebra of $\mathrm{SO}(n)$ are anti-symmetric, which, in all generality, constitute only a subset of all possible transformations in $\mathrm{GL}(n)$. Nevertheless, we obtain again in the first order:

$$\mathbf{x}_{t+1}\mathbf{x}_t^\top - \mathbf{x}_t\mathbf{x}_{t+1}^\top = (\mathbf{x}_t + \theta\mathbf{A}\mathbf{x}_t)\mathbf{x}_t^\top - \mathbf{x}_t(\mathbf{x}_t + \theta\mathbf{A}\mathbf{x}_t)^\top$$
$$= \theta\mathbf{A}\mathbf{x}_t\mathbf{x}_t^\top - (\theta\mathbf{A}\mathbf{x}_t\mathbf{x}_t^\top)^\top \ . \tag{6}$$

## B  K-means and PCA for transformation learning

We proposed to evaluate the SM and NSM model for transformation learning in relation with associated K-means and PCA models for transformation learning. The models are respectively defined as the solution of the following optimization problems:

$$\min_{\boldsymbol{A}\in\mathbb{R}^{K\times n^2}} \sum_t \|\chi_t - \boldsymbol{A}^\top \boldsymbol{A}\chi_t\|^2 \quad \text{s.t.} \quad \boldsymbol{A}\boldsymbol{A}^\top = I \ , \tag{7}$$

$$\min_{\boldsymbol{\Theta}\in\mathbb{R}_+^{K\times T}, \boldsymbol{A}_{a,:}^\top \in\mathbb{R}^{n^2}} \sum_{t,a} \Theta_{t,a} \|\chi_t - \boldsymbol{A}_{a,:}^\top\|^2 \quad \text{s.t.} \quad \sum_{a=1}^{K}\Theta_{t,a} = 1 \quad \forall t\in\{1,\ldots,T\} \ . \tag{8}$$

## C Detecting motion by correlating spatial and temporal derivatives

In the following we consider the approximation of the learned filters Fig.2a by the cartoon version Fig. 2b (main text). The magnitude of translation can be evaluated by solving a linear regression between the temporal, $\Delta_t \mathbf{x}_t$, and spatial, $\Delta_i \mathbf{x}_t$, derivatives of pixel intensities. Conventionally, this is done by minimizing the mismatch squared,

$$\min_{\theta_t} \|\Delta_t \mathbf{x}_t + \theta_t \Delta_i \mathbf{x}_t\|^2 + \lambda \theta_t^2 \,, \tag{9}$$

where the first term enforces object constancy [7] and the second second term is a regularizer which may be thought to arise from a Gaussian prior on the velocity estimator. By differentiating (9) with respect to $\theta_t$ and setting the derivative to zero, we find:

$$\theta_t = \Delta_t \mathbf{x}_t^\top \Delta_i \mathbf{x}_t / (\|\Delta_i \mathbf{x}_t\|^2 + \lambda) \approx \Delta_t \mathbf{x}_t^\top \Delta_i \mathbf{x}_t / \lambda \,. \tag{10}$$

The latter approximation is justified by the fact that the regularizer $\lambda$ dominates the denominator in the realistic setting [8]. This demonstrates that the magnitude of translation per frame is given by the normalized correlation of the spatial and temporal derivatives across the visual field.

The spatial gradient of pixel intensity, $\Delta_i \mathbf{x}_{t,i}$ at pixel $i$ can be computed as the mean of the differences with the right and the left nearest pixels [7] :

$$\Delta_i \mathbf{x}_{t,i} = \frac{1}{2}((\mathbf{x}_{t,i+1} - \mathbf{x}_{t,i}) + (\mathbf{x}_{t,i} - \mathbf{x}_{t,i-1})) = \frac{1}{2}(\mathbf{x}_{t,i+1} - \mathbf{x}_{t,i-1}) = [\mathbf{A}\mathbf{x}_t]_i \,, \tag{11}$$

where matrix, $\mathbf{A}$, matches the generator of translation learned by the various models above.

$$\mathbf{A} = \frac{1}{2} \begin{pmatrix} 0 & +1 & 0 & 0 & 0 \\ -1 & 0 & +1 & 0 & 0 \\ 0 & -1 & 0 & +1 & 0 \\ 0 & 0 & -1 & 0 & +1 \\ 0 & 0 & 0 & -1 & 0 \end{pmatrix} \,. \tag{12}$$

Then, the magnitude of translation, $\theta_t$, is proportional to:

$$\Delta_t \mathbf{x}_t^\top \Delta_i \mathbf{x}_t = \Delta_t \mathbf{x}_t^\top \mathbf{A} \mathbf{x}_t = \begin{pmatrix} \vdots \\ \Delta_t \mathbf{x}_{i-1,t} \\ \Delta_t \mathbf{x}_{i,t} \\ \Delta_t \mathbf{x}_{i+1,t} \\ \vdots \end{pmatrix}^\top \begin{pmatrix} \vdots \\ \mathbf{x}_{t,i} - \mathbf{x}_{t,i-2} \\ \mathbf{x}_{t,i+1} - \mathbf{x}_{t,i-1} \\ \mathbf{x}_{t,i+2} - \mathbf{x}_{t,i} \\ \vdots \end{pmatrix} \,. \tag{13}$$

## D Learning rotations and translations of 2D images

In addition to learning 1D translations of 1D images, SM and NSM can learn other types of transformations.

### D.1 Planar rotations of 2D images

We applied our model to pairs of randomly generated two-dimensional images rotated by a small angle relative to each other. To this end, we first generated seed images with random pixel intensities. From each seed image, we generated a transformed image by applying small clockwise and counterclockwise rotations with different angles. We then presented these pairs to the algorithms. Again, we chose $K = 2$, and the models were evaluated against standard PCA and K-means as described in the main text Section 3.

NSM applied to rotated $5 \times 5$ pixel images learns transformation matrix of clockwise and counterclockwise rotations, Fig.1a and 1b. Similarly, K-means recovers both generators of rotations (results not shown). As before, SM and PCA recover only one rotation generator. In their work Rao et al. [5] could also also only account for one direction of rotation, either clockwise or counter-clockwise as it is the case for SM and PCA.

Figure 1: Learning and evaluation of rotations of 2D images. The filters learned by NSM (a) accounting for clockwise rotation are displayed as an array of weights that, for each pixel of $\Delta \mathbf{x}_t \in \mathbb{R}^{5 \times 5}$, shows the strength of its connection to each of the $\mathbf{x}_t$'s matrix pixels. Similarly for (b) the filters accounting for counterclockwise rotations. In (c) the learned filter accounting for clockwise rotation is applied multiple times to diagonal bar (read left to right, top to bottom).

A naive evaluation of the accuracy of the learned filters was performed by applying a learned filter to a diagonal bar on a $5 \times 5$ pixel image as shown in Fig. 1c. After multiple application of the rotation operator, artifacts start to appear. We can here observe the limitations of the Lie algebra generators instead of the Lie groups and exponential maps, which would account for large transformations.

Reshaping the filters shows that each component of the filter connects $\Delta \mathbf{x}_{t,i,j}$ only to nearby $\mathbf{x}_{t,i\pm1,j\pm1}$ and the connection between $\Delta \mathbf{x}_{t,i,j}$ and $\mathbf{x}_{t,i,j}$ is absent as before. This generalizes the three-pixel model to another type of transformation. It plausibly explain how the Drosophila circuit responsible for detecting roll, pitch and yaw [2] is learned.

## D.2  Translations of 2D images

In the case of 1D translations and 2D rotations there is only one generator of transformation (for sign-unconstrained output), which explains the choice of $K = 2$ for signed-constrained output. In the case of 2D images undergoing both horizontal and vertical motions our model learns two different generators, left-right and up-down motion ($K = 4$ for sign-constrained output). Fig.2a-b-c-d show the filters learned by our model, each accounting for a motion in a cardinal direction. These generators were also reported in [4].

Figure 2: Learning of translations of 2D images. The filters learned by NSM (a) accounting for right-to-left horizontal motion are displayed as an array of weights such that, for each pixel of $\Delta \mathbf{x}_t \in \mathbb{R}^{5 \times 5}$, shows the strength of its connection to each of the $\mathbf{x}_t$'s matrix pixels. Similarly for (b) the filters accounting for left-to-right horizontal, (c) downward and (d) upward motions.

## E  Equivalence of Global Motion Estimators

Using the expression for translation magnitude derived in the previous section we can show that the integration of the three-pixel model output over the visual field produces the same result as that of the integration of the output of the popular Hassenstein-Reichardt detector (HRD) [6]. This is done by considering the discretized time derivative $\Delta_t \mathbf{x}_t = \mathbf{x}_t - \mathbf{x}_{t-\tau}$, where $\tau$ would identify as a

time-delay in HRD. The following holds true when the learned filters Fig.2a are approximated by the cartoon version Fig. 2b (main text).

Consider first the central pixel $i$ for which the temporal derivative is taken. The output activity of our detector, $y_i(t)$, can be obtained as follows:

$$y_i(t) = \big(x_i(t) - x_i(t-\tau)\big) \times \big(x_{i+1}(t) - x_{i-1}(t)\big) \tag{14}$$
$$= x_i(t)x_{i+1}(t) - x_i(t)x_{i-1}(t)$$
$$-x_i(t-\tau)x_{i+1}(t) + x_i(t-\tau)x_{i-1}(t) \tag{15}$$

Consider now pixel $i+1$ as central. Then

$$y_{i+1}(t) = \big(x_{i+1}(t) - x_{i+1}(t-\tau)\big) \times \big(x_{i+2}(t) - x_i(t)\big)$$
$$= x_{i+1}(t)x_{i+2}(t) - x_{i+1}(t)x_i(t) - x_{i+1}(t-\tau)x_{i+2}(t) + x_{i+1}(t-\tau)x_i(t)$$

Then after adding $y_i(t)$ and $y_{i+1}(t)$ given above, the terms depending on $(t)$ but not on $(t-\tau)$ cancel each other. The other terms, with a dependence in $(t-\tau)$ can be combined to produce HR detectors leading to the following

$$y_i(t) + y_{i+1}(t) = -x_i(t)x_{i-1}(t) + x_{i+1}(t)x_{i+2}(t)$$
$$+x_i(t-\tau)x_{i-1}(t)$$
$$+x_{i+1}(t-\tau)x_i(t) - x_i(t-\tau)x_{i+1}(t)$$
$$-x_{i+1}(t-\tau)x_{i+2}(t).$$

By denoting $\mathbf{HR}(i,t) = x_{i+1}(t-\tau)x_i(t) - x_i(t-\tau)x_{i+1}(t)$ and by adding successive elements one can obtain the following:

$$\sum_{i=-n}^{n} y_i(t) = \sum_{i=-n}^{n-1} \mathbf{HR}(i,t)$$
$$+x_{-n-1}(t-\tau)x_{-n-1}(t) - x_n(t-\tau)x_{n+1}(t)$$
$$-x_{-n}(t)x_{-n-1}(t) + x_n(t)x_{n+1}(t) \tag{16}$$

This proves a formal equivalence between the proposed model and the HRD when averaged over the pixels of the visual field. See illustration in Fig.3, since for large fields the boundary contribution vanishes.

Figure 3: Equivalence between (a) HRD and (b) the cartoon version of the learned model and after integration over the visual field

Interestingly, the expression (16) can be evaluated by the neural circuit suggested by fly connectomics. The visual field is tiled by EMD neurons each EMD neuron, $i$, computing a product between $\Delta_t x_{i,t}$ and $A_{i,:}\mathbf{x}_t$. Then the outputs of the EMD neurons throughout the visual field are summed in a giant neuron signaling global motion.

Therefore, the calculation in (13) can account for the three arms of the neural EMD, suggesting that each EMD neuron receives the temporal derivative of the middle pixel light intensity and multiplies it by the difference of the light intensity between the left- and the right- nearest pixel.