[Reviews · NeurIPS 2019]

Reviewer 1



This paper examines neural circuits for motion detection, which have been extensively modeled and are certainly of interest to the field. A nice connection is drawn with recent work on biologically plausible learning of similarity matching objective functions, which is also of great interest to the field. While the work draws a nice connection on timely topics, some of the conclusions and biological predictions were difficult for me to follow. While the paper is decently written, the presentation is dense and difficult to follow in several places. This is partly due to space constraints on the manuscript, but I nonetheless think there is room for improvement in sections 2.1, section 3, and section 4. Section 2.1 and the preceding paragraph would be easily understood by readers with a background in group theory (I imagine this would be mostly limited to physicists), but I don't think this level of sophistication is necessary to convey the main ideas of the paper. It should certainly be mentioned that Lie groups provide an elegant perspective, but I think this is currently over-emphasized, and jargon terms like "generators" are over-used, without much explanation. In reality the model is simply learning something pretty simple due to the linearity assumption on line 113. A greater effort should be made to provide readers with intuition, e.g., "the model aims to learn a small set of canonical, linear transformations A_1, A_2, etc., which may, for example, denote rotations or translations of input images. Formally, these may be viewed as a Lie Algebra..." Section 3 is a bit long and unclear. I think the main takeaway is that the model learns reasonable transformations corresponding to shifts and rotations. I am unsure why the comparisons to PCA and K-means are important. Perhaps this section could be shortened/streamlined? Perhaps it does not deserve its own section and should be absorbed into section 2 or 4. Section 4 is maybe the most confusing to me, and could be clarified. Lines 256-262 describe a model very informally (no equations) which would require the reader to do a lot of work to parse and understand. It is not 100% clear to me how things like back-propagating action potentials could support the Hebbian learning rule without interfering with signals flowing from synapse to soma. Making more explicit references to equation 18 would help here (it took me a while to figure out that this was the relevant expression).

Reviewer 2



As far as I know, the model is novel. As was recognized earlier by Rao & Ruderman, Lie groups are an elegant way to think about visual invariance, and this model appears to be the first attempt to specify how the brain might learn to approximate such a representation. One thing that I felt was lacking was a discussion of why Lie groups are useful at a conceptual level. The authors offer a very brief technical introduction and then jump right into the learning model. It's significance as a biological model is a bit hard for me to ascertain, because the authors don't really evaluate it with respect to benchmark physiological data, apart from a few stylized facts. What one would want to see is that known receptive field properties of these neurons emerge from this learning rule. For example, in their discussion of mammalian pyramidal cells, the authors argue that known properties of these neurons would allow them to carry out these computations, but I didn't see any discussion of how the representations posited by their model offer a better account of cortical tuning properties than any alternative model. In the discussion of the Hassenstein-Riechardt detector, it was interesting to see that the model could capture the dependence on three pixels (although this confused me because neurons don't get information about pixels in the real world). But I would have liked to see that the tuning functions for this model actually better account for fly motion detector tuning functions compared to the Hassenstein-Reichardt detector, or any other model (surely there are more recent ones?). Minor comments: p. 5: "Now we have identified" -> "Now that we have identified" ------------ Post-rebuttal comments: The responses to my comments were thoughtful. In terms of model comparison, I was looking for comparisons in terms of how well the models fit data (what are the data that this model explains and other models don't). It didn't seem like the response provided that. Instead, they showed that it was more robust compared to another model, which doesn't say anything about fit to empirical data. Overall, I've decided to increase my rating by one point.

Reviewer 3



Quality: I found the notion of similarity preserving transformations interesting, as well the posited learning rules. In terms of normative modeling, I found it somewhat odd that the authors considered only the case of K=2 (Line 212) filters. Given a reasonable set of natural movie statistics, it would be interesting to see what filters are generated by larger values of K, and whether they correspond to those seen in fly vision. Clarity: I found the paper very dense to read and difficult to understand. While some of this is necessary because of the mathematical machinery involved, I have the sense that the clarity could be improved by putting some of the intuition first. For example, after going through all the math of section 2, one of the core results is given by Figure 2. After thinking it through, I found the core idea that translations are good at approximating future images from past images very intuitive, but during the process of reading the paper and working through the math, very little of that intuition came across. This might be a matter of taste, but I’d consider myself a fairly mathematical person and I still had a lot of trouble working my way through all the formalism. Originality and Significance: I would classify this work as original, and possibly significant. Receptive fields forming through a combination of local learning rules and natural image is an interesting area of research, and the authors make some progress towards positing plausible objective functions as well as local learning rules

[Author Response · NeurIPS 2019]

We thank the reviewers for their feedback. Below we list the respective revisions that will be made (**R = Reviewer**).

**Conceptual and intuitive introduction to transformation learning (R1,R2,R3)** We will completely revise Section 2 which will now open with the following paragraph: "Next, we turn our attention to learning to detect transformations from pairs of consecutive video frames. We start with the observation that much of the change in pixel intensities in consecutive frames arises from a local translation of the image. For small translations pixel intensity change is given by a linear operator (or matrix) multiplying the vector of pixel intensity scaled by the magnitude of translation. Because, for a 2D image, multiple directions of translation are possible, there is a set of translation matrices with corresponding magnitudes. Our goal is to learn both the translation matrices from pairs of consecutive video frames and the magnitudes of translations for each pair. Such a learning problem will reduce to the one discussed in the previous section, but performed on an unusual feature – the outer product of pixel intensity and variation of pixel intensity vectors."

**Results of learning in our model (R1,R3)** We will revise Section 3. Specifically, we present PCA and K-means results because these are well understood computations that help with an intuitive understanding of our biologically plausible algorithm. PCA illustrates the learning of generators in the sign-unconstrained case and K-means illustrates the effect of constraining the sign of the output.

In the case of 1D translations and 2D rotations there is only one generator of transformation (for sign-unconstrained output), which explains the choice of $K = 2$ for signed-constrained output. In the case of 2D images undergoing both horizontal and vertical motions our model learns two different generators, left-right and up-down motion ($K = 4$ for sign-constrained output). Fig.1c-d-e-f show the filters learned by our model, each accounting for a motion in a cardinal direction. These generators were also reported in [17]. In addition, when presented with pairs of points in $\mathbb{R}^n$ transformed by the elements of group $SO(n)$, our model learns the various generators ($K > 2$).

**Comparison of model predictions with the biological observations (R1,R2,R3)** Our theory's predictions are consistent with experimental measurements of physiology and anatomy of the T4 circuit including phi and reverse phi optical illusions. The predicted output of our detectors, integrated over the visual field is consistent with experimental observations such as the increase with image contrast, the oscillations in the motion signal locked to the phase of the visual stimulus, non-monotonic dependence of output on motion velocity. Our reference to pixels in the context of fly vision is justified by the facet structure of the fly eye wherein photoreceptors respond to light intensity in a hexagonal grid of locations in the visual field.

**Biological implementation of the algorithm (R1,R2)** We will revise Section 4 to clarify the relevant biological mechanisms and make a stronger connection with the algorithm. In particular, it is true that backpropagating action potential briefly interrupts dendritic integration yet it is widely thought to underlie Hebbian-like learning [32].

**Comparison of our model with other models (R1,R2)** The main difference between our model and most published models (including the model in ref.[28]) is that the motion detector is learned from data using biologically plausible learning rules in an unsupervised setting. Thus, our model can generate somewhat different receptive fields for different natural image statistics such as that in ON and OFF pathways potentially accounting for minor differences reported between T4 and T5 circuits [33]. In addition, the model in [28] is architecturally different from ours as it is composed of a shared non-delay line flanked by two delay lines. Our model instead uses a temporal derivative in the middle pixel flanked with two non-shared non-delay lines. Whereas, after integration over the visual field, the outputs predicted by our model, HR and [28] are algebraically the same, the predicted output of a single motion detector in our model is different from both HR and [28].

Figure 1: Robustness to noise (a) our model vs HR, (b) our model vs [28]. Learned generators on 2D images of (c-d) horizontal motions and (e-f) vertical motions.

A very recent paper [1] reported experimental measurements of direction opponency (DO) in T4 and T5 cells. They showed that the HR model cannot account for DO and proposed a biophysical model that reproduces observed DO. Our model also reproduces DO, as will be demonstrated in the revised version of our paper.

Finally, we evaluated our model against HR and [28] in terms of robustness of their output to noise. Fig.1a (resp.1b) show the relative difference in mean squared error (MSE) between our model and HR (resp.[28]), for different SNR and different number of detectors. A positive value indicates that our model is less sensitive to noise than the competition. For both low SNR (<0dB) and integration over a large number of detectors our model, HR, and [28] perform similarly. In realistic settings, however, our model is more robust to noise than the other two.

[1] Bara A. Badwan et al. Dynamic nonlinearities enable direction opponency in drosophila elementary motion detectors. *Nature neuroscience*, 22(8):1318, 2019.


[Meta-Review · NeurIPS 2019]

This work considers similarity-preserving objective functions for learning to classify inputs with a temporal dimension. The authors propose a modification of the Lie algebra formulation of Ruderman and Rao, where the algorithm maximizes the similarity of transformation of inputs that are nearby in time rather than comparing inputs at the same time directly. While the scores given were worthy of acceptance, the enthusiasm of reviewers both in the body of the reviews and in the discussion was somewhat muted. My impression is that there were two main reasons for this. a) the difference between the proposed approach and competing accounts (e.g. in Salazar-Gatzimas et al.) is not explained sufficiently, making it difficult to assess novelty (although this has been addressed in the rebuttal, and that material should be moved to the paper itself) and b) the extent to which this model accounts for natural data better than other models (as indicated by a pure goodness-of-fit measure or prediction accuracy rather than robustness to noise, or theoretical arguments) is unclear. Thus, while I see no reason to contradict the recommendation of the reviewers that the paper be accepted, we expect the reviewers to address these points (and the clarity of the paper in general) in the camera ready version of the paper.